# When does risk outweigh reward? Identifying potential scoring strategies with netball's new two-point rule

**Aaron S. Fox** *, **Lyndell Bruce**

Centre for Sport Research, School of Exercise and Nutrition Sciences, Deakin University, Melbourne, Victoria, Australia

* aaron.f@deakin.edu.au

**Data Availability Statement:** The data and code that can be used to replicate the analysis and results are available at https://zenodo.org/record/3941375#.Xwvh7igza70.

## Abstract

Changing rules to promote scoring through more 'high-risk' play has become common in team sports. Australia's national netball league (i.e. Suncorp Super Netball) has recently taken this approach–introducing a two-point shooting rule. Teams will be awarded two-points for shots made from an 'outer circle' 3.0m-4.9m from the goal in the final five minutes of quarters. We sought to answer a series of questions regarding the implementation and potential strategies surrounding the two-point rule in Suncorp Super Netball. We used video coded data from the 2018 Suncorp Super Netball season to identify the total number of made and missed shots from different distances across the season. We also used shooting statistics from recent Fast5 Netball World Series (a competition with a two-point shooting rule already in place) as a comparator. The reward of two-points is relatively well-aligned to the relative risk of missing shots from the proposed outer versus inner circle (2.22 [1.98, 2.48 95% CIs]) based on existing shooting data from Suncorp Super Netball teams. We found that the relative risk of missing shots from 'long-' (i.e. 3.5m-4.0m) versus 'mid-range' (i.e. 3.0–3.5m) was only slightly elevated (1.52 [1.21, 1.86 95% CIs])–suggesting teams should favour long- over mid-range shots when the two-point shot is available. Based on the typical number of shots a team receives in a five-minute period, we found that teams may be able to score ~3.51 extra points per quarter when taking all versus no-shots from the two-point outer circle. Analysis of the Fast5 versus Suncorp Super Netball data did, however, reveal that shooting accuracy from long-range may decrease when a two-point shot is available. Teams may need to consider situational factors (e.g. altered opposition defensive strategies) when developing their shooting strategy for taking advantage of the two-point shot.

## Introduction

Netball is a court-based team sport played among Commonwealth countries, and has one of the highest participation rates for team sports in Australia [1]. Like many court-based team sports, the aim of netball is to score more points than the opposition team. Netball is, however, unique in that goals may only be scored by two players on each team from within a 'shooting

**Funding:** The author(s) received no specific funding for this work.

**Competing interests:** The authors have declared that no competing interests exist.

circle' (i.e. a half circle with a 4.9m radius) at their end of the court [2]. In standard netball formats, goals scored from within this circle result in one point for the team [2]. Akin to other sports (such as cricket's T20 format), netball has developed a 'fast-paced' version of the game–labelled as 'Fast5'–which includes different scoring rules [2]. One point is still awarded for successful shots within an 'inner circle' 3.5m from the goal [2]. However, two points are awarded for a shot within an 'outer circle' 3.5m-4.9m from the goal; while three points are awarded for a shot from outside both circles (i.e. > 4.9m) [2].

Until recently, these modified scoring rules have stayed in the adapted form of the game. Australia's national elite-level netball league (i.e. Suncorp Super Netball) has, however, made the decision to introduce a form of the two-point 'outer circle' rule in the 2020 season [3]. Teams will be awarded with two-points for shots made from an outer circle 3.0m-4.9m from the goal, but only in the final five minutes of quarters (Fig 1) [3]. The rule change–announced two months prior to the start of the season–has elicited strong public opinions from players and fans, often in opposition to the proposed changes [4–7].

Changing rules to promote scoring, or direct changes to scoring rules are not uncommon to team sports. The International Cricket Council (ICC) introduced the 'power-play' to one-day cricket in 2005 –adding periods of play where certain fielding restrictions are in place that potentially promote more risky high scoring play. The National Football League (NFL) introduced the choice of a two-point conversion option following touchdowns in 1994 –an option to run an additional on-field play from the two-yard line to reach the end zone for two points versus the standard place kick from 15-yards (or from 2-yards prior to 2015). This allows teams to potentially increase the value of a touchdown, but at a likely higher degree of difficulty (i.e. success rate has typically fluctuated around 50% across years) compared to the traditional one-point conversion (i.e. success rate typically remains over 90% across years) option. The National Basketball Association (NBA) adopted a three-point line in 1979, providing players an opportunity to score an extra point for a shot attempted from a greater distance. In all of these cases, the greater scoring opportunities provided by the rule change come with an element of added risk. Teams and players must consider whether a play with higher risk is worth the added value of the extra points.

Altered scoring rules in sport can provide teams an opportunity to adjust their tactical approach and be a catalyst for a change in game style. Perhaps the best example of this is the shift in play style towards a greater number of three-point attempts seen in the NBA in recent years. When introduced in 1979, teams took 2.8 three-point shots on average–which dramatically contrasts to the 32.0 three-point shots taken per game in the most recent 2018–19 season [8]. This increase appears to have come at the cost of the 'mid-range' shot. In the early 2000s, mid-range shots made up ~35% of a team's total shots versus ~18% from the three-point line [8]. This ratio has effectively flipped, with now ~15% versus ~35% of total shots coming from mid-range versus three-point, respectively [8]. It is unknown why a lag in predominant uptake of the three-point shot exists from when the rule was introduced in 1979 to post 2000; nor is their direct evidence to support why teams have dramatically shifted away from mid-range to three-point shots in recent years. However, we speculate that the increased reliance on statistics and analytics for informing decision-making in sport may be a factor in this modern era shift. The shift away from mid-range to three-point shots appears driven by: (i) three-points is worth more than two-points (*obviously*. . .); but perhaps more importantly, (ii) the added value of an extra point outweighs the added risk of the shot from further out. In simple terms, the three-point shot provides a better 'risk-reward' trade-off.

The parallel between the three-point line in basketball to the new two-point rule being introduced in netball is easy to see. Despite the two-point shot only being available in the final five-minutes of each 15-minute quarter, its presence may provide teams with a significant

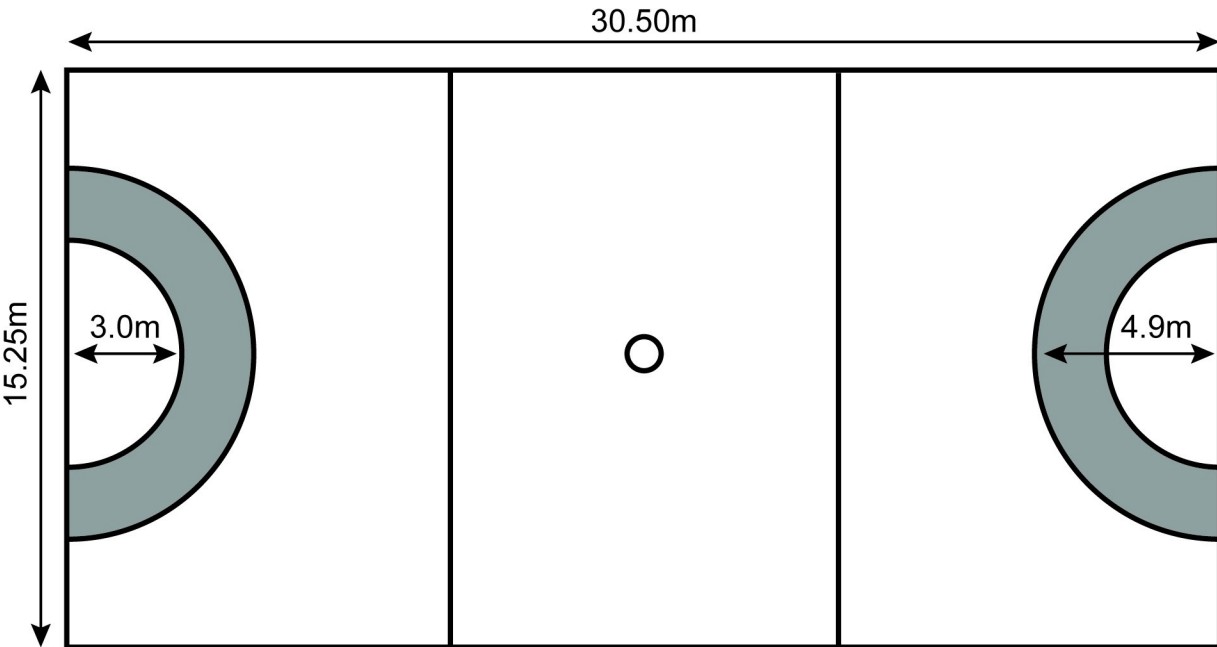

**Fig 1. Netball court schematic with the new 'outer-circle' (highlighted in grey) where a two-point shot will be available for the final five minutes of quarters.**

consideration in their tactical approach to the match. In this study we aim to answer a series of questions regarding the implementation and potential use of the two-point rule in the 2020 Suncorp Super Netball season, specifically: (i) is the weighting of a 2:1 value appropriate based on the relative risk of missing a shot from greater versus less than 3m from goal; (ii) should teams consider minimising the use of 'mid-range' shots during the five-minute two-point scoring period; (iii) how much added value does the two-point rule offer, and theoretically what proportion of shots from the two-point 'outer circle' may maximise a team's scoring opportunity during the five minute period where the shot is available; and (iv) compare shooting success between Fast5 and Suncorp Super Netball to examine how the situational context of having a two-point shot available (i.e. as in Fast5) might impact shooting success.

## Materials and methods

### Participants

Participants for this study included all players across the eight teams from the 2018 season of the Australian national netball league (i.e. Suncorp Super Netball) and all players across the 2016 and 2017 Fast5 Netball World Series. An exemption from ethics review (and subsequent waiver of individual consent) was granted by the Deakin University Human Research Ethics Committee (id: 2018–255).

### Data collection

Broadcast video footage was obtained from all regular season matches (*n* = 56; 14 rounds x 4 matches) across the 2018 Suncorp Super Netball season. Video footage was coded by a single individual using the NetballStats application ('app') (Perana Sports, Australia) on an iPad (iOS 10, Apple Inc., Australia). NetballStats is designed to capture every possession in netball, including the specific location of each possession as an XY coordinate on the court. During

coding–the coder was able to pause, rewind and play the video as required, to capture all instances of play. Possessions were coded based on the team and playing position receiving the ball, the activity being performed related to the possession (e.g. pass, circle entry, shot on goal), and the location on the court where the ball was received. After each match was coded, the raw file code and the related *.csv* file with all coded events were exported. Although all match events were coded, the present study extracted only the shot on goal events (both made and missed shots) from the dataset. The XY coordinates of each shot were used to calculate the distance the shot was taken from the goal. The NetballStats app codes the XY coordinate on a scale of 0–100 and 0–200 for court width and length, respectively. Based on the standard netball court dimensions of 15.25m (width) and 30.5m (length), each coordinate unit was calculated as 0.1525m. The goal posts were assumed at the XY coordinates of [50,0] and [50,200] for each end. The radial distance in the coded XY coordinate from the designated goal end was then multiplied by the coordinate unit (i.e. 0.1525m) to determine the distance the shot was taken from goal.

We also used shooting statistics from International Fast5 matches ($n$ = 36) across the 2016 and 2017 Fast5 Netball World Series in this study. This competition comprises the top six ranked netball nations who participate in the Fast5 format. The main rules changes for Fast5 (versus Suncorp Super Netball) include six-minute quarters (versus the standard 15 minutes), only five players are on court (goal shooter, goal attack, centre, goal defence and goal keeper positions), rolling substitutes, two- (shots from 3.0 to 4.9m) and three-point (shots from outside the 4.9m shooting circle) shots, centre passes going to the non-scoring team, and a 'power-play' where each team chooses one quarter in which all their goals will be worth double [2]. These data were manually extracted from the Champion Data match centre, who are the official provider of statistics for the competition. Specifically, we extracted the total number of made and missed shots (including those in regular play and penalty shots) for the one- and two-point areas across all matches from all teams.

## Data analysis

Our study required estimating the probability of missing shots from various distances (i.e. distance zones or 'bins') on the court. We achieved this by defining a beta distribution in a probability density function for different distance bins, specified by:

$$f(x, a, b) = \frac{\Gamma(a+b)x^{a-1}(1-x)^{b-1}}{\Gamma(a)\Gamma(b)}$$

where $a$ and $b$ represent the number of missed and made shots within a distance bin, respectively; $x$ is the probability of $a$ relative to $b$; and $\Gamma$ is the gamma function [9,10]. Probability density functions were created for made and missed shots in the following distance bins and datasets: (i) 0.0m to 3.0m (i.e. proposed new 'inner circle') and greater than 3.0m (i.e. proposed new 'outer circle') in the Suncorp Super Netball data; (ii) distance bins at 0.5m intervals from the goal post in the Suncorp Super Netball data; (iii) less than 3.5m (i.e. Fast5 one-point area) and greater than 3.5m (i.e. Fast5 two-point area) in the Suncorp Super Netball and Fast5 data. We used the Monte Carlo approach to randomly sample values ($n$ = 100,000) from the probability density functions to gather the probability of missing shots from the various distance bins; and subsequently used these values to answer our studies research questions.

To determine the appropriateness of the new 2:1 point ratio in Suncorp Super Netball for shots from the inner versus outer circle, we compared the average relative odds (± 95% confidence intervals [CI]) of missing from the outer versus inner circle. This was achieved by dividing the randomly sampled values from the probability density functions of the outer by those

from the inner circle at each sample iteration. Theoretically, the relative odds of missing from the outer to inner circle should match the ratio of points awarded (i.e. 2:1) for the new scoring rule to be appropriate.

To understand whether close- or mid-range shots are still a viable approach when long-range two-point shots are on offer–we compared the average relative odds (± 95% confidence intervals [CI]) of missing shots from the different 0.5m interval distance bins from the goal post. There are no distinct boundaries as to what could be considered 'close-' or 'mid-range' within the netball shooting circle, hence why we chose to examine distance bins at 0.5m intervals. Contrasting these relative risks with the reward of two- versus one-point provided guidance on whether shots from further versus closer distance bins are a risk worth taking. We interpreted shots from the distance bins within the inner circle (i.e. < 3.0m) where the relative odds of missing compared to distance bins in the outer circle (i.e. > 3.0m) dropped below two (i.e. the added value of shooting from the outer circle) as less 'valuable' (i.e. the reward of making shots from further out outweighs the reduced risk of missing from closer in).

To determine how much added value the two-point rule may offer, we considered two scenarios. First–we calculated the proportion of shots taken from the respective 0.5m distance bins across a typical five-minute period by averaging the total number of shots from these distances across the entire duration of the Suncorp Super Netball season (i.e. 56 matches x 60 minutes each). We considered both the proportion of shots and the probability of success from the 0.5m distance bins from our randomly sampled values, and calculated the average point increase a team would expect with the two-point value added to shots from the outer circle (i.e. > 3.0m). Second–we estimated the number of points a team could gain with different proportions of shots from the inner and outer circle during a typical five-minute period. We tested shooting proportions in the outer circle from 0% to 100%, at 10% increments. Based on the previously calculated probabilities of making versus missing shots from within and outside 3.0m –we estimated the number of points a team would score during a typical five-minute period with these different distributions.

To determine whether the situational context of having a two-point shot available would impact shooting success–we compared similar shooting statistics from Suncorp Super Netball (i.e. no two-point shot) to International Fast5 (i.e. two-point shot available) matches. Specifically, we compared the relative odds of missing from within the Fast5 one- and two-point zones in Fast5 versus Suncorp Super Netball competition. The use of the 3.5m threshold is slightly different to the proposed two-point rule for Suncorp Super Netball, however this distance represents where one- versus two-point shots are awarded in Fast5 competition [2]. This therefore provides a better comparison for how situational context may affect shooting success when scoring rules are changed. We used identical methods to those previously outlined to examine the relative risk of missing shots from within and outside 3.5m in Fast5 versus Suncorp Super Netball matches. Contrasting the relative risks between the two competitions provided insight on whether there are different probabilities of missing shots from similar shooting distances when the scoring rules are different.

The data and code that can be used to replicate the analysis and results are available at https://zenodo.org/record/3941375#.Xwvh7igza70.

## Results

We observed a total of 7,487 shots over the 56 matches of the Suncorp Super Netball season. Of these, 6,585 and 902 were taken from the proposed inner (i.e. < 3.0m) versus outer (i.e. > 3.0m) circle, respectively. The number of successful shots were 5,875 (success rate of 89.22%) and 686 (success rate of 76.05%) from the inner and outer circles, respectively. Random

**Table 1. Total number of made, missed, total shots and proportion (%) of all shots, and the success rate from distance bins at 0.5m intervals from the goal post.**

| Distance Bin | Made Shots | Missed Shots | Total Shots | % Shots | Success Rate |
|---|---|---|---|---|---|
| 0.0m – 0.5m | 107 | 6 | 113 | 1.51% | 94.69% |
| 0.5m – 1.0m | 522 | 30 | 552 | 7.37% | 94.57% |
| 1.0m – 1.5m | 1,521 | 113 | 1,634 | 21.82% | 93.08% |
| 1.5m – 2.0m | 1,845 | 187 | 2,032 | 27.14% | 90.80% |
| 2.0m – 2.5m | 1,143 | 192 | 1,335 | 17.83% | 85.62% |
| 2.5m – 3.0m | 737 | 182 | 919 | 12.27% | 80.20% |
| 3.0m – 3.5m | 476 | 124 | 600 | 8.01% | 79.33% |
| 3.5m – 4.0m | 144 | 65 | 209 | 2.79% | 68.90% |
| 4.0m – 4.5m | 62 | 24 | 86 | 1.15% | 72.09% |
| 4.5m – 4.9m | 4 | 3 | 7 | 0.09% | 57.14% |

sampling from the probability density functions created from these data revealed that the relative odds of missing from the outer versus inner circle was 2.22 [1.98, 2.48 95% CI].

The number of made, missed and total shots from the 0.5m interval distance bins are listed in Table 1. We observed a consistent increase in the risk of missing shots as distance from the goal increased; however, the relative odds of missing from adjacent distance bins were often similar (i.e. 95% CIs were close to or overlapped 1) (Table 2).

The typical number of shots in a five minute period for one team was ~5.57 (i.e. 7,487 total shots divided by 56 games x 60 minutes and two teams). Teams took a small proportion (i.e. 12.05%) of shots from greater than 3.0m from the goal. Based on the shot proportions from each distance bin (see Table 1) and our random sampling–in a typical five-minute period teams would score 4.88 ± 0.02 versus 5.39 ± 0.03 (mean ± SD) points with the standard versus new two-point scoring rules, respectively. Our simulated data of teams taking an increasing proportion of shots from the new outer circle (i.e. > 3.0m) revealed a progressive decrease and increase in the points scored under the standard and two-point scoring rules (Fig 2).

**Table 2. Relative odds [± 95% confidence intervals] of missing shots as distance increases.** Each column compares the relative odds of missing compared to the current row.

| | 0.0m-0.5m | 0.5m-1.0m | 1.0m-1.5m | 1.5m-2.0m | 2.0m-2.5m | 2.5m-3.0m | 3.0m-3.5m | 3.5m-4.0m | 4.0m-4.5m | 4.5m-4.9m |
|---|---|---|---|---|---|---|---|---|---|---|
| 0.0m-0.5m | — | 1.21 [0.54, 2.39] | 1.55 [0.73, 2.97] | 2.06 [0.99, 3.94] | 3.21 [1.55, 6.14] | 4.43 [2.13, 8.48] | 4.62 [2.21, 8.86] | 6.95 [3.28, 13.36] | 6.24 [2.79, 12.25] | 9.58 [2.61, 21.33] |
| 0.5m-1.0m | — | — | 1.31 [0.93, 1.8] | 1.75 [1.26, 2.37] | 2.73 [1.98, 3.71] | 3.76 [2.72, 5.1] | 3.93 [2.81, 5.35] | 5.91 [4.14, 8.17] | 5.30 [3.4, 7.74] | 8.14 [2.79, 14.77] |
| 1.0m-1.5m | — | — | — | 1.34 [1.1, 1.61] | 2.10 [1.73, 2.51] | 2.89 [2.38, 3.46] | 3.01 [2.45, 3.65] | 4.53 [3.57, 5.63] | 4.07 [2.86, 5.46] | 6.25 [2.22, 10.79] |
| 1.5m-2.0m | — | — | — | — | 1.57 [1.33, 1.83] | 2.16 [1.84, 2.52] | 2.26 [1.88, 2.67] | 3.40 [2.74, 4.12] | 3.05 [2.17, 4.03] | 4.68 [1.67, 8.04] |
| 2.0m-2.5m | — | — | — | — | — | 1.38 [1.18, 1.61] | 1.44 [1.21, 1.70] | 2.17 [1.75, 2.63] | 1.95 [1.39, 2.57] | 2.99 [1.07, 5.13] |
| 2.5m-3.0m | — | — | — | — | — | — | 1.05 [0.88, 1.24] | 1.58 [1.27, 1.91] | 1.42 [1.01, 1.87] | 2.17 [0.78, 3.72] |
| 3.0m-3.5m | — | — | — | — | — | — | — | 1.52 [1.21, 1.86] | 1.36 [0.96, 1.81] | 2.09 [0.75, 3.6] |
| 3.5m-4.0m | — | — | — | — | — | — | — | — | 0.91 [0.63, 1.23] | 1.39 [0.49, 2.42] |
| 4.0m-4.5m | — | — | — | — | — | — | — | — | — | 1.58 [0.54, 2.86] |

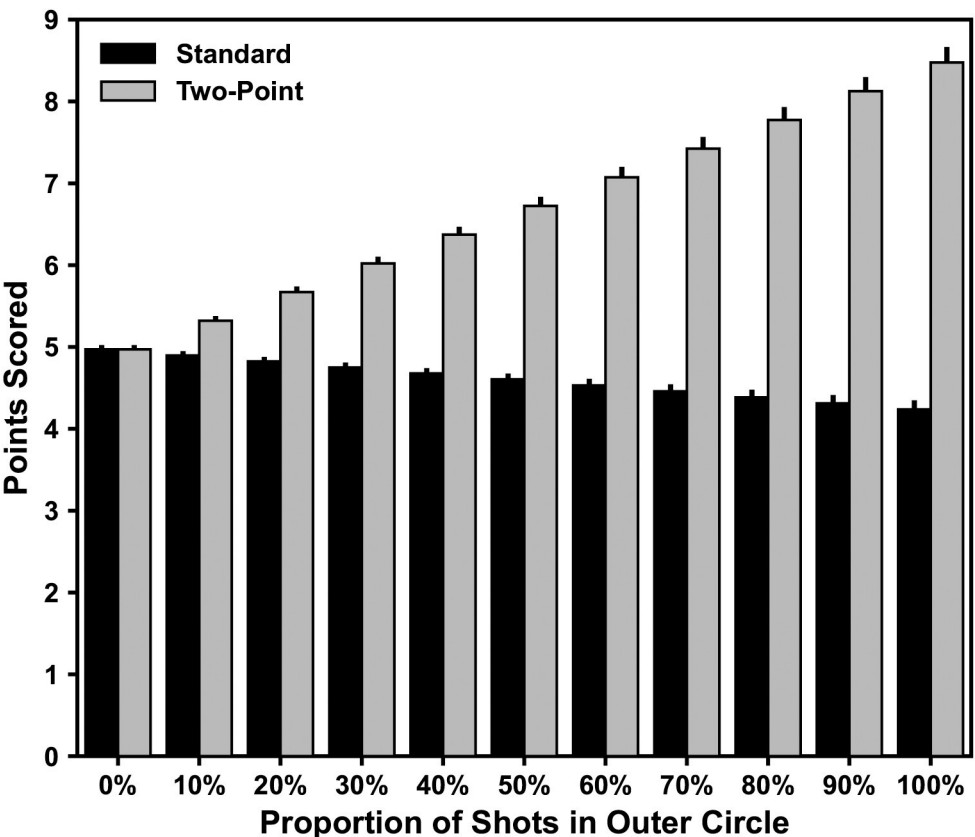

**Fig 2. Simulated number of points scored with different shooting proportions from the 'outer circle' during a typical five-minute period with standard (i.e. one-point for all shots) versus the two-point (i.e. two-points from the outer circle) rule systems.**

When considering the less versus greater than 3.5m distance bins (i.e. Fast5 one- vs. two-point shot)– 7,185 and 302 shots were taken, respectively, from these bins in the Suncorp Super Netball data. The number of successful shots were 6,351 (success rate of 88.39%) and 210 (success rate of 69.54%) from within and outside of 3.5m, respectively. In the Fast5 International data–of the 1,822 total shots (excluding three-point shots), 787 and 1,035 were taken from the within and outside of 3.5m, respectively. The number of successful shots were 664 (success rate of 84.37%) and 367 (success rate of 35.46%) from within and outside of 3.5m, respectively. The relative odds of missing in Fast5 versus Suncorp Super Netball were 1.35 [1.16, 1.55 95% CI] and 2.14 [1.84, 2.48 95% CI] from within and outside of 3.5m, respectively.

## Discussion

This study sought to understand the implementation and potential use of the new two-point rule introduced for the 2020 Suncorp Super Netball. We found that the 2:1 point ratio appears relatively appropriate given the risk of missing from within versus outside of the new 3.0m outer circle. Our data suggests that teams should consider minimising the use of 'mid-range' shots (i.e. shots from 2.5m-3.0m) in favour of 'longer-range' (e.g. 3.0m-3.5m) when the two-point rule is in effect, given the reward of 2:1 points outweighs the elevated risk of missing from this slightly greater distance. In addition, our simulated data suggests teams may gain the most points from taking as many shots as possible from the outer circle during the five-minute

period when the two-point shot is available. However, examining a competition where a two-point rule was in effect (i.e. Fast5) revealed that the presence of the two-point rule could elevate the risk of missing shots from this longer distance.

## Is a 2:1 point ratio appropriate?

Our results found that the relative odds of missing from the new outer versus inner circle was 2.22 [1.98, 2.48 95% CIs]. This finding suggests that the 2:1 point ratio for the outer to inner circle is relatively appropriate. On average, players would be taking a shot attempt that is 2.22 times riskier of missing–with a two times higher reward. The two-point value is, however, towards the low end of the 95% confidence intervals–so there could still be contention as to whether the two-point shot represents 'good' value. Our results do suggest that awarding three-points (or more) to shots from the outer circle would be over-inflating its value. Altogether, when simply considering the two zones–it appears that the risk of the longer shot does closely match the reward. This presents a new tactical consideration for attacking teams, in that shots from 'long-range' may now become a valid or appropriate choice. The two-point rule may also require tactical adjustment by defensive players. Defenders often position themselves to force their opponent away from the goal post and into taking shots from further out as a means to reduce accuracy. With greater points on offer for shots from further out, teams may need to reconsider this defensive strategy.

## Should teams abandon close- or mid-range shots?

The introduction of a 'long-range' scoring shot with added value has seen tactical adjustments in basketball–perhaps the most appropriate comparative sport to netball. The dramatic increase in three-point shots in basketball over recent years appears to have been at the demise of the 'mid-range' shot [8]. Our findings suggest a similar phenomenon could become apparent in netball. We found that the risk of missing shots from distances in the two-point outer circle (i.e. > 3.0m) can become dramatically higher when compared to shots taken close to the goal post (i.e. < 1.5m) (Table 2). For distances greater than 3.0m from the goal post, the risk of missing shots is three times higher or greater when compared to shots taken within 1.5m. In this context, the reward of twice as many points does not seem to outweigh the elevated risk of missing from the added distance. Shots very close to goal (i.e. 'close-range') may therefore still be appropriate due to the low-risk involved, even when the two-point shot is available. The value of taking shots from the outer-circle becomes more appropriate as shot distance extends past 1.5m. Compared to the 1.5m to 2.0m range, the relative risk of missing a shot from 3.0m to 3.5m was 2.26 [1.88, 2.67 95% CI]. This elevated risk is more closely aligned with the 2:1 point reward associated with the longer shot–so could therefore be considered as a calculated risk. Extending beyond 3.5m when compared to 1.5m to 2.0m does, however, elevate the risk of missing beyond what could be considered appropriate (i.e. > 3 times the risk of missing).

An important finding from our results is that there appears to be little value in attempting shots from between 2.0m to 3.0m when two-points are on offer for shots from 3.0m to 3.5m. The reward of double points appears to outweigh the added risk for extending shot distance from 2.0m-2.5m to 3.0m-3.5m (1.44 [1.21, 1.70 95% CI]). Further, shots from 2.5m to 3.0m have an almost identical risk of missing as shots from 3.0m to 3.5m (1.05 [0.88, 1.24 95% CI]). Eliminating shots from 2.0m to 3.0m from the goal post when the two-point rule is in play may be appropriate. Compared to the 2.0m to 3.0m distance bins, at no point does the relative risk of missing a shot from greater than 3.0m dramatically outweigh the 2:1 point ratio on offer. If more points are on offer at an appropriately weighted or even similar risk, why should players take the lower value shot? Our findings suggest this is a relevant question when the

two-point rule is in play in netball–and that 'mid-range' (i.e. 2.0m to 3.0m) shots should be minimised in favour of shots in the outer circle.

Our analysis also provides some important tactical considerations around a team's line-up formations and attacking strategies when the two-point shot is available. It appears there is 'safe' value in taking close- versus long-range shots, while there is better value in taking long- versus mid-range shots. The combination of the two shooting positions (i.e. Goal Shooter and Goal Attack) may therefore represent a key consideration during the two-point shot period. Teams may benefit from using a combination of a tall 'holding' post shooter (i.e. remaining close to goal) with someone who is more proficient from long-range, as a means to take advantage of shots from the two varying distances. This combination may have the added benefit of providing a tall rebounder, increasing the chance of second shot opportunities from long-range misses. Teams may also need to adapt their attacking strategies in how they move the ball around and into the shooting circle, and how their two shooters position within the circle to promote shot opportunities from both close- and long-range. These practical considerations are also likely relevant to teams from a defensive perspective–whereby changes to defensive positions in line-ups and defensive strategies will require consideration in response to their opponent.

The aforementioned tactical considerations are focused on the performance or defence of the two-point shot. However, netball can be viewed as a complex system with many integrating parts [11]–and one rule change may have a 'knock-on' effect across the sport. We highlight the need to consider shooting circle line-ups, yet teams may also require varied mid-court line-ups including players more effective at generating long-range shots from circle feeds. The introduction of a new scoring rule could also impact on the overall make-up, and how perceived value is distributed across a team. Player's expressed concerns over the introduction of the two-point rule, mostly due to a lack of consultation, but also surrounding how a rule that potentially boosts the value of shooting positions could impact on the value of defensive players. If teams perceive a need to attract shooting players through high-value contracts, this may lower the league-wide value of defensive players and impact their earning potential through the sport. Player discontent, in general, may also lead to other various unintended negative consequences. Our study was focused on assessing performance strategies related to the new two-point rule, hence understanding these potential systemic effects are outside the scope of our work. The broader impact of such singular rule changes must, however, be considered in future research.

These tactical considerations are focused on two point shot. . .however it is important to note the complexities of netball as a sport. . .and how changing one rule could create multiple changes around it. . .knock-on effect/systemic influences. . .changes to mid-court players who are potentially better at opening up circle feeds for long-range shots. . .potential increase in value for shooters and the concern that this may make defenders less valuable (see Jo Weston article). . .

## What proportion of shots should be taken from the outer circle?

Our analyses revealed that teams currently take a small proportion (i.e. 12.05%) of shots from greater than 3.0m from the goal. This is not surprising, as shots closer to goal are easier and there is no scoring benefit to shooting from a greater distance. The addition of the two-point rule does, however, provide teams a choice about the tactical approach they can take with this new rule. Specifically, how much they should shift their shooting proportion to the outer circle when two-points are on offer. It is quite plausible that teams will dramatically increase their proportion of shots from greater than 3.0m when the two-point shot is on offer. The 12.05%

rate from a past season presented in our study provides a baseline comparator for future work examining the 2020 (and potential future) Super Netball seasons.

Previously, we suggested that teams could still consider 'close-range' shots (i.e. within 1.5m) given the much lower risk associated with these attempts. On this alone, you could conclude that some proportion of a team's shots should still be taken from in close even when the two-point shot is on offer. This approach, however, does not consider the typical number of shots available to a team in the five minute period where two-points are on offer. Further, even though a team would likely miss more shots with more shots from the outer circle–it does not consider whether this would still result in more points. We calculated that if teams kept the same proportion of shots in the outer circle that is currently typical, they would only receive a small benefit (i.e. 4.88 ± 0.02 vs. 5.39 ± 0.03) with the two-point rule.

When simulating increasing proportions of shots taken from the outer circle, we observed a progressive increase in the typical points scored over a five-minute period–peaking at 8.48 ± 0.16 when 100% of shots are taken from the outer circle. This represents an increase of ~3.51 points with the dramatic shift (i.e. 0% to 100%) of shots taken from the outer circle (Fig 2). The explanation for the change in points scored comes back to the risk versus reward relationship of shots taken from the outer circle. If we examine when all shots are taken from the inner circle, players will be highly accurate–making approximately 89.23% of these shots (i.e. on average almost 5 out of the 5.57 available). When all shots are taken from the outer circle, players will be less accurate–making approximately 76.12% of these shots (i.e. on average just above 4 out of the 5.57 available). The major difference here is that the drop-off in shot success is easily made up for by having double the points available. Our data suggests that teams would only need to make at least three long-range shots of the typical five to six shots available to make the 100% outer circle approach worthwhile.

This approach does, however, still present high risk versus reward and may be impacted by many situational factors. First–given that teams can only expect to get five or six shots during the two-point period, having a slightly 'off' period and missing two or three extra shots from long-range could result in less points scored than a relatively safer approach. Second–teams could find themselves in a slower-paced match and receive fewer shots than usual in the five-minute period. This may become prevalent if teams take longer to set-up two-point scoring opportunities during the five-minute period. In these cases, the decision to risk two-points and missing versus taking an easier one-point shot carries a much heavier weight. Third–having a player in a 'shooting slump' and subsequently missing a number of shots during the five-minute period could be quite impactful given the few opportunities available. There is mixed evidence to support the notion of the 'cold-' or 'hot-hand' in shooting sports [12]. Nonetheless, the introduction of 'rolling' substitutions to the coming Suncorp Super Netball season [3] would provide teams an opportunity to immediately substitute out a 'cold' shooter. There is, however, no guarantee that their replacement could immediately come off the bench 'hot.' Fourth–added situational or defensive pressure associated with a highly valuable shot could influence the probability of success in a negative manner [13,14]. All of the aforementioned situational factors were not considered within the simulated approach we took. Nonetheless, this analysis provides some support for the tactical approach of teams taking as many shots as possible from the outer circle when two-points are on offer for these shots.

## Could availability of the two-point shot impact success?

Our analysis of the Fast5 versus Suncorp Super Netball data reveals the potential effect of situational factors on long-range shooting success. We found that the probability of successful shot attempts from comparative distances is reduced in the Fast5 format–but the magnitude of this

effect is dependent on the distance from goal. The probability of missing shots from within 3.5m was slightly elevated (1.35 [1.16, 1.55 95% CI]) in Fast5 compared to the Suncorp Super Netball data. The slight increase could be due to a number of reasons. The shorter quarters and associated breaks, combined with fewer players on the court results in Fast5 being a more open and faster paced netball variant. This pace and associated fatigue could have an impact on a players shooting ability. The lack of a Wing Attack player in Fast5 [2] would also impact on the structure of play relating to shooting circle entries and shot opportunities. With one less player to 'feed' the shooting circle, the attacking team may find it difficult to generate shooting opportunities very close to the goal.

Our results did not account for where within the 3.5m distance the shots were taken from between the competitions. It is possible that more shots within 3.5m in the Fast5 matches were from a longer distance, which would impact accuracy. A much more dramatic effect on shot success was found between Fast5 and Suncorp Super Netball matches when examining shots from greater than 3.5m (2.14 [1.84, 2.84 95% CI]). The same factors that we discussed relating to shots within 3.5m could have generated this reduction in shot success. An additional explanation for this large increase could be that opposing teams may have altered their defensive approach to long-range shots in Fast5 matches. In standard scoring formats, defensive players may sacrifice their defensive position on a long-range shot to better position themselves to rebound, and more so rely on the difficulty of the long-range shot to cause a miss. The added value of the two-point shot in the Fast5 format may have required defensive players to re-think this strategy–and resulted in more defensive pressure being applied to the long-range shot.

The presence of a 'power-play' in specific quarters during Fast5 (i.e. double points for goals) may have amplified this–where defensive teams may have further elevated their pressure to prevent four-point shots being made. These reasons are, however, speculative–as we cannot infer the defensive pressure or match situation a shooting player was under during the shot attempts we analysed. Additional research analysing the match situation (e.g. proximity of defenders) and relation to shot success could yield further details around why long-range shot success differs between Fast5 and the more traditional format. Whatever the reason may be, these data provide additional context as to whether the 2:1 scoring ratio of the new outer circle is an attractive option. Our earlier analysis suggested that the relative risk of missing shots from this distance was appropriately balanced with the additional point reward. However, if the risk of missing from this range is further elevated by situational factors when two-point shots are available (as was the case in the Fast5 format)–the risk of missing could actually outweigh the reward.

An additional finding from the Fast5 data was that the proportion of 'long-range' shots increased compared to the Suncorp Super Netball data. Only 4.20% of shots were taken from greater than 3.5m in Suncorp Super Netball matches, compared to 56.81% of shots in Fast5 matches (not accounting for additional three-point shots). This appears to represent a significant tactical shift in the attacking priorities of teams when additional points are available for long-range shots. It is plausible that we will see a similar shift in Suncorp Super Netball with the two-point rule in place, as well as any other netball competitions that adapt rules in a similar way. If the scoring rules of netball continue to be adapted in this manner–long range shooting ability may become a much more important skill for netball shooters, potentially resulting in a need to alter training and player development strategies for shooting positions.

## Limitations

There are certain limitations to consider when interpreting our results. First–we took a general approach to understanding typical shooting behaviour and success across a single season of

the Suncorp Super Netball league. This did not take into account different team behaviour or individual performance statistics. Teams may be better or worse off using a long-range shooting strategy depending on the ability of their shooting personnel to make shots from long-range.

Second–the use of a past seasons data meant that the shots we examined were from matches where standard scoring was in place. It is plausible that opposition defensive strategy will change when two-points are available, potentially impacting the accuracy of shots from the outer circle in a similar fashion to what we observed in Fast5 matches. Similarly, the reasoning for players taking 'long-range' shots (e.g. time constraints, reacting to opposition defensive strategies vs. specifically positioning for long range shots) is likely to change with the two-point rule in place. This is another likely reason for the differences we observed between Super Netball and Fast5 data, and will be an important consideration in assessing data from the 2020 Super Netball season and beyond. Understanding the impact of the new two-point rule on team/player shooting behaviours and success is an important area for future investigations.

Third–having only one seasons worth of data resulted in a limited number of shots from the furthest distance bins (i.e. > 4.0m). Additional data would strengthen the probability functions we used to model accuracy from these distances.

Fourth–our analysis did not take into account how the timing in the quarter and potential fatigue may impact shooting success. Shots from the outer circle will be awarded two points in the final five minutes of a quarter [3]. If fatigue had an impact on a players shooting accuracy, particularly from long-range–teams may need to reconsider a shooting strategy emphasising long-range shots at the end of the quarter. There is mixed evidence that physical fatigue negatively impacts accuracy and sports-based skill tasks [15–17]–with none of this literature relating to netball. The ability to now substitute players immediately inside quarters [3] and having 'fresh' shooting players on the court may negate any potential effect of fatigue on shooting accuracy at the end of quarters.

Lastly–our results are only applicable to netball competitions where two-point (or more) shooting rules are implemented (i.e. Fast5 and Suncorp Super Netball). Other competitions may adopt similar rule changes in the future, and our findings may become more widely applicable at a later date.

## Conclusions

Our findings provide some support that teams should take advantage of the new two-point rule during the five-minute period it is in play. We have provided evidence that teams may be able to score approximately three extra points in each five-minute period by adopting an outer circle focused shooting strategy (i.e. taking 100% of shots from this area)–equating to a potential 12 point increase across the entire match. Given netball matches can often be decided by a few points or less, taking advantage of this tactical approach could mean the difference between winning versus losing. However, the capacity to increase scoring may change if situational factors (e.g. defensive strategies) incur an elevated risk of missing long-range shots during periods where the two-point shot is available.

Not surprisingly, our data demonstrates that the best odds of scoring from the outer circle is to take shots from as close to the inner edge as possible (i.e. 3.0m-3.5m). The use of 'mid-range' shots towards the outside of the inner circle (i.e. 2.0m-3.0m) when two-point shots are available is not supported by our data–where we found the reward associated with shifting these shots back to the outer circle outweighed the elevated risk of missing. Teams should consider designing plays that result in circle entries and shot attempts from as close to 3.0m out as possible as part of their attacking strategy. If players find themselves in a shooting position

towards the outside of the inner circle, a strategy of feeding the ball out and repositioning to the outer circle is also likely to be advantageous. From a defensive perspective, teams likely need to adapt their strategies against long-range shots to limit their opposition's use of the two-point shot.

## Acknowledgments

The authors would like to thank Jeff Fleming for their assistance with data collection.

## Author Contributions

**Conceptualization:** Aaron S. Fox, Lyndell Bruce.

**Data curation:** Aaron S. Fox, Lyndell Bruce.

**Formal analysis:** Aaron S. Fox.

**Investigation:** Aaron S. Fox, Lyndell Bruce.

**Methodology:** Aaron S. Fox, Lyndell Bruce.

**Project administration:** Aaron S. Fox, Lyndell Bruce.

**Writing – original draft:** Aaron S. Fox, Lyndell Bruce.

**Writing – review & editing:** Aaron S. Fox, Lyndell Bruce.

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
