## [Decision Letter · Decision Letter 0]

17 Sep 2020

PONE-D-20-21619

When Does Risk Outweigh Reward? Identifying Potential Scoring Strategies with Netballs New Two-Point Rule

PLOS ONE

Dear Dr. Fox,

Thank you for submitting your manuscript to PLOS ONE. After careful consideration, we feel that it has merit but does not fully meet PLOS ONE’s publication criteria as it currently stands. Therefore, we invite you to submit a revised version of the manuscript that addresses the points raised during the review process.

The revision needs to address the methodological concerns relating to the data set and the design. In addition, the limitations of the work need to further highlighted in the discussion. Please respond in a point by point manner to the reviewers comments.

We look forward to receiving your revised manuscript.

Kind regards,

Caroline Sunderland

Academic Editor

PLOS ONE

Journal Requirements:

Reviewers' comments:

Reviewer's Responses to Questions

**Comments to the Author**

1. Is the manuscript technically sound, and do the data support the conclusions?

Reviewer #1: No

Reviewer #2: Yes

2. Has the statistical analysis been performed appropriately and rigorously? 

Reviewer #1: Yes

Reviewer #2: Yes

3. Have the authors made all data underlying the findings in their manuscript fully available?

Reviewer #1: Yes

Reviewer #2: Yes

4. Is the manuscript presented in an intelligible fashion and written in standard English?

Reviewer #1: Yes

Reviewer #2: Yes

5. Review Comments to the Author

Reviewer #1: Overall, the topic is interesting and worthy of scientific attention. From a scientific perspective the data analysis was performed well.

My major concern is that the 2018 SSN data analysed was from prior to the two-point rule being implemented. The motivation and tactical strategy for players to shoot from outer edge (3 m to 4.9 m) of the circle was not present in 2018. The two-point rule would presumably change the tactics and subsequent behaviours of players. As such, the results are not representative of the current rule change. Furthermore, the Fast5 is not representative of either the old or new rules, given the 3.5 m threshold, less assisting players and faster pace. While these are acknowledged as limitations, I think the data does not truly support the Discussion.

Other questions I have include:

How does the 12.05% of shots from greater than 3 m compare to now, when the two-point rule is in place?

Were the shots from 3 m to 4.9 m in the 2018 SSN season made because of time pressures, defensive structures, or were they genuine attempts at goal as they would be with the new scoring rule? It is difficult to know.

I think some practical considerations should be discussed to highlight how the new rule may influence the way SSN is played. For example, how have tactics changed and how will this influence team selection (i.e. a good close shooter and a good long-range shooter). This will no doubt influence how teams now recruit players as well.

The implementation of the two-point scoring rule is an interesting topic and has potential for numerous future research applications. On top of the already discussed practical considerations are, obtaining data about players behaviours both qualitative and quantitative, how does this influence international matches that have traditional scoring methods, how the shooting from long range changes as a function of match status etc.

I believe the study would be better suited to a comparison of a SSN season with the old scoring rule and the new scoring rule. As it is, it is quite speculative, and could potentially have a negative influence on match tactics given the data does not represent the actual state-of -play.

Reviewer #2: Thank you for the opportunity to review this paper which was very well presented and very topical within the Australian netball scene. Whilst this may seem like a narrow focus given the rule change is related to a domestic competition, the fact that Australia is one of the world’s leading netball nations suggests that rule changes such as this could also be incorporated in future international standard rules competitions as they are currently in the shorter version of Fast5 netball. This paper has used data from recent years to develop probabilities and possibilities associated with goal shooting and scoring under the new rules. Whilst there are many factors in a game that will influence the potential scoring strategies and thus probable outcomes as shown in this paper, it has provided a very good starting point. No doubt future studies will be able to assess actual strategies used and tactics developed in response to this rule change, however it is important in the early days to have some data to inform practice.

Introduction

introduction provides a good background to the study and an explanation of the rules and current practice

line 66-67 - it is not clear why there might be a higher degree of difficulty compared to the traditional one point conversion

lines 81-87 - interesting comparison to the introduction of the three point shot in basketball and the increasing use of three point shots in today’s game. The authors have suggested why the three point shot is more popular however these same reasons would have been applicable at all times. Is it possible to suggest why it has been more used since 2000?

Lines 91-99 - the authors state that the aim of the paper is to answer a series of questions regarding implementation and potential use of the two-point rule. I would suggest that question one and two are retained as research questions that can be addressed with the data provided, however, I would recommend that RQ3 and 4 are reworded and refined to reduce the subjectivity and more closely align to the data that is available i.e., remove “should” and “could”; RQ4 – make reference to the comparison between SSN and Fast5.

Materials and methods

line 113 - each match was coded using the relevant broadcast footage - this could be deleted as line 108 refers to the broadcast video footage

line 125 - please confirm whether it was the radial distance that was calculated

How many games were analysed from the Fast5 format?

Did the goals analysed include penalty shots?

Data analysis

line 178 - suggest inclusion of reference to Suncorp Super Netball

The methodology used to answer questions three and four provides an interesting perspective.

Results

Lines 223-227 – this is really interesting data and I would have found it easier if it was included in Table 1.

Discussion

By the very nature of the data and the aim of the paper the discussion is very speculative but it does present a range of possible scenarios which demonstrates a good understanding of the game and these are acknowledged as being speculative as you cannot guess what will happen in the game. Likewise the limitations and conclusions are appropriately stated.

Editorial

very long paragraphs or lack of throughout the paper - these could be reviewed

Abstract Line 17 – though should be through

6. PLOS authors have the option to publish the peer review history of their article (what does this mean?). If published, this will include your full peer review and any attached files.

Reviewer #1: No

Reviewer #2: No

---

## [Author Response · Author response to Decision Letter 0]

6 Oct 2020

Please see document attached with submission for detailed response to reviewer and editorial comments.

---

## [Decision Letter · Decision Letter 1]

29 Oct 2020

PONE-D-20-21619R1

When Does Risk Outweigh Reward? Identifying Potential Scoring Strategies with Netball's New Two-Point Rule

PLOS ONE

Dear Dr. Fox,

Thank you for submitting your manuscript to PLOS ONE. After careful consideration, we feel that it has merit but does not fully meet PLOS ONE’s publication criteria as it currently stands. Therefore, we invite you to submit a revised version of the manuscript that addresses the points raised during the review process.

Please review and respond to the minor amendments suggested by the reviewers.

We look forward to receiving your revised manuscript.

Kind regards,

Caroline Sunderland

Academic Editor

PLOS ONE

Reviewers' comments:

Reviewer's Responses to Questions

**Comments to the Author**

1. If the authors have adequately addressed your comments raised in a previous round of review and you feel that this manuscript is now acceptable for publication, you may indicate that here to bypass the “Comments to the Author” section, enter your conflict of interest statement in the “Confidential to Editor” section, and submit your "Accept" recommendation.

Reviewer #1: All comments have been addressed

Reviewer #2: (No Response)

2. Is the manuscript technically sound, and do the data support the conclusions?

Reviewer #1: Yes

Reviewer #2: Yes

3. Has the statistical analysis been performed appropriately and rigorously? 

Reviewer #1: Yes

Reviewer #2: Yes

4. Have the authors made all data underlying the findings in their manuscript fully available?

Reviewer #1: Yes

Reviewer #2: Yes

5. Is the manuscript presented in an intelligible fashion and written in standard English?

Reviewer #1: Yes

Reviewer #2: Yes

6. Review Comments to the Author

Reviewer #1: The authors have done a good job responding to the comments. Well done. This is an interesting first step, hopefully the authors continue this line of research given that there is now a full season of data with the rule change.

I do have one more comment that I think could strengthen the article, and have a broader message for sport in general.

In the changes made in response to reviewer comment C1.4. there is an opportunity to highlight the complexity inherent within netball and team sports in general. For example, how one rule change will create multiple emergent properties that come about from it. For instance where you now talk about the influence of the new rule on tactics, substitutions, and strategy, through to recruitment etc, I think the message needs to be conveyed that changing one rule will have knock on systemic influences and that need to be considered.

For example, I think about the 6, 6, 6 rule introduced to AFL and the negative unintended consequences that came about due to this rule change. Also, I remember the backlash from some high-profile netball coaches and players regarding the lack of consultation about the rule changes. Not all changes will have the intended effect, and consideration of negative consequences needs be front of mind for sport stakeholders.

The following reference can be used to support this argument about complexity of netball.

Mclean S, Hulme A, Read, G, Mooney M, Bedford A, Salmon, P. (2019) A systems approach to performance analysis in women’s netball: Using Work Domain Analysis to model elite netball performance. Frontiers in Psychology. 10:201. DOI: 10.3389/fpsyg.2019.00201.

Reviewer #2: Thank you for your considered responses which have added little extra detail to the paper. I think the rewording of the research questions is much better.

With respect to my comment R2.3 I was referring more to the difference between the statistics reported from early 2000 until now, rather than the lag between 1979 to post 2000, and as such I don’t think this additional information in lines 84-86 is necessary as it is not clearly stated what the percentage was at 1979, so comparisons cannot be made. However it is stated that from early 2000’s to now, the make-up of the shots has changed from 35% mid-range shots and 18% three-point shots, to then flip to 15% mid-range and 35% three-points. My question was why has this flip occurred as the reasons for using the three-point shot as described by the authors would have also been relevant in the early 2000’s or in fact when it was first introduced …..so why now (or over the years) has this increased use of the three-point shot developed? It was a slow change maybe….so is that what we will expect in netball?

I am not sure anyone would have a definitive answer unless there has been a change in something else, but it is likely to be just an increased ability of players to shoot the 3 pointer, or a change in the acceptance of risk in taking these shots maybe? Unless the authors have a different response I would suggest removing the additional comment as stated above (lines 84-86), and I am happy to leave the content as it was originally.

Line 114 and 119 - I think there was a misunderstanding with my comment, this insertion is not necessary as it is clearly stated on line 114 that broadcast video footage was used and on lines 115 and 116 that this video footage was coded by a single individual. I would suggest the removal of this new sentence.

7. PLOS authors have the option to publish the peer review history of their article (what does this mean?). If published, this will include your full peer review and any attached files.

Reviewer #1: No

Reviewer #2: No

---

## [Author Response · Author response to Decision Letter 1]

29 Oct 2020

Please see document attached with submission for detailed responses to reviewer comments.

---

## [Editor Report · Decision Letter 2]

9 Nov 2020

When Does Risk Outweigh Reward? Identifying Potential Scoring Strategies with Netball's New Two-Point Rule

PONE-D-20-21619R2

Dear Dr. Fox,

We’re pleased to inform you that your manuscript has been judged scientifically suitable for publication and will be formally accepted for publication once it meets all outstanding technical requirements.

Kind regards,

Caroline Sunderland

Academic Editor

PLOS ONE
---

## [Editor Report · Acceptance letter]

12 Nov 2020

PONE-D-20-21619R2 

When Does Risk Outweigh Reward? Identifying Potential Scoring Strategies with Netball’s New Two-Point Rule 

Dear Dr. Fox:

I'm pleased to inform you that your manuscript has been deemed suitable for publication in PLOS ONE. Congratulations! Your manuscript is now with our production department. 

Kind regards, 

on behalf of

Dr. Caroline Sunderland 

Academic Editor

PLOS ONE